# CONSTRAINED BAYESIAN OPTIMIZATION FOR AUTOMATIC CHEMICAL DESIGN

## ABSTRACT

Automatic Chemical Design provides a framework for generating novel molecules with optimized molecular properties. The current model suffers from the pathology that it tends to produce invalid molecular structures. By reformulating the search procedure as a constrained Bayesian optimization problem, we showcase improvements in both the validity and quality of the generated molecules. We demonstrate that the model consistently produces novel molecules ranking above the 90th percentile of the distribution over training set scores across a range of objective functions. Importantly, our method suffers no degradation in the complexity or the diversity of the generated molecules.

## 1 INTRODUCTION

There are two fundamental ways in which Machine Learning can be leveraged in chemical design:

1. To evaluate a molecule for a given application.
2. To find a promising molecule for a given application.

There has been much progress in the first use-case through the development of quantitative structure activity relationship (QSAR) models using deep learning (Ma et al., 2015). These models have achieved state-of-the-art results in predicting properties of known molecules.

The second use-case, finding new molecules that are useful for a given application, is arguably more important however. One existing approach for finding molecules that maximise an application-specific metric involves searching a large library of compounds, either physically or virtually Pyzer-Knapp et al. (2015). This has the disadvantage that the search is not open-ended; if the molecule is not in the library you specify, the search won't find it.

A second method involves the use of genetic algorithms. In this approach, a known molecule acts as a seed and a local search is performed over a discrete space of molecules. Although these methods have enjoyed success in producing biologically active compounds, an approach featuring a search over an open-ended, continuous space would be beneficial. The use of geometrical cues such as gradients to guide the search in continuous space could accelerate both drug (Pyzer-Knapp et al., 2015; Gómez-Bombarelli et al., 2016a) and material (Hachmann et al., 2011; 2014) discovery by functioning as a high-throughput virtual screen of unpromising candidates.

Recently, Gómez-Bombarelli et al. (Gómez-Bombarelli et al., 2016b) presented Automatic Chemical Design, a VAE architecture capable of encoding continuous representations of molecules. In continuous latent space, gradient-based optimization is leveraged to find molecules that maximize a design metric.

Although a strong proof of concept, Automatic Chemical Design possesses a deficiency in so far as it fails to generate a high proportion of valid molecular structures. Gómez-Bombarelli et al. (2016b) hypothesise that molecules selected by Bayesian Optimization lie in "dead regions" of the latent space far away from any data that the VAE has seen in training, yielding invalid structures when decoded. Although there have been many attempts to address the issue of generating valid molecules, they all rely on model assumptions that can negatively impact either the complexity (Jaques et al., 2017; Janz et al., 2017; Kusner et al., 2017) or the diversity (Jin et al., 2018) of the generated molecules.

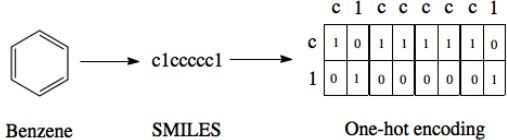

Figure 1: The SMILES and one-hot encoding for benzene. For simplicity only the characters present in Benzene are shown in the one-hot encoding. In reality there would be a column for each character in the SMILES alphabet.

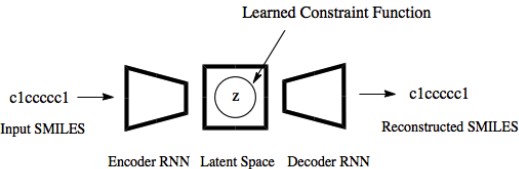

Figure 2: The SMILES Variational Autoencoder with the learned constraint function illustrated by a circular feasible region in the latent space.

The principle contribution of this paper will be to present an approach based on constrained Bayesian optimization that generates a high proportion of valid sequences without relying on model assumptions that affect the complexity or the diversity of the generated molecules.

## 2 METHODS

### 2.1 SMILES REPRESENTATION

SMILES strings Weininger (1988) are a means of representing molecules as a character sequence. This text-based format facilitates the use of tools from natural language processing for applications such as organic chemistry reaction prediction Schwaller et al. (2017); Jin et al. (2017). For use with the variational autoencoder, the SMILES strings in turn are converted into one-hot vectors indicating the presence or absence of a particular character within a sequence as illustrated in Figure 1.

### 2.2 VARIATIONAL AUTOENCODER

Variational autoencoders Kingma & Welling (2013); Kingma et al. (2014) allow us to map molecules $\mathbf{x}$ to and from continuous values $\mathbf{z}$ in a latent space. The encoding $\mathbf{z}$ is interpreted as a latent variable in a probabilistic generative model over which there is a prior distribution $\mathbf{p}(\mathbf{z})$. The probabilistic decoder is defined by the likelihood function $p_\theta(\mathbf{x}|\mathbf{z})$. The posterior distribution $p_\theta(\mathbf{z}|\mathbf{x})$ is interpreted as the probabilistic encoder. The parameters of the likelihood $p_\theta(\mathbf{x}|\mathbf{z})$ as well as the parameters of the approximate posterior distribution $q_\phi(\mathbf{z}|\mathbf{x})$ are learned by maximizing the evidence lower bound (ELBO)

$$\mathcal{L}(\phi, \theta; \mathbf{x}) = \mathbb{E}_{q_\phi(\mathbf{z}|\mathbf{x})}[\log p_\theta(\mathbf{x}, \mathbf{z}) - \log q_\phi(\mathbf{z}|\mathbf{x})].$$

Variational autoencoders have been coupled with recurrent neural networks by Bowman et al. (2015) to encode sentences into a continuous latent space. This approach is followed for the SMILES format both by Gómez-Bombarelli et al. (2016b) and here. The SMILES variational autoencoder, together with our constraint function, is shown in Figure 2.

## 2.3 OBJECTIVE FUNCTIONS FOR BAYESIAN OPTIMIZATION OF MOLECULES

Bayesian Optimization is performed in the latent space of the variational autoencoder in order to find molecules that score highly under a specified objective function. We assess molecular quality on the following objectives:

$$J_{\text{comp}}^{\text{logP}}(m) = \text{logP}(m) - \text{SA}(m) - \text{ring-penalty}(m),$$

$$J_{\text{comp}}^{\text{QED}}(m) = \text{QED}(m) - \text{SA}(m) - \text{ring-penalty}(m),$$

$$J^{\text{QED}}(m) = \text{QED}(m).$$

$m$ denotes a molecule, $\text{logP}(m)$ is the water-octanol partition coefficient, $\text{QED}(m)$ is the quantitative estimate of drug-likeness (Bickerton et al., 2012) and $\text{SA}(m)$ is the synthetic accessibility (Ertl & Schuffenhauer, 2009). The ring penalty term is as featured in (Gómez-Bombarelli et al., 2016b). The "comp" subscript is designed to indicate that the objective function is a composite of standalone metrics.

It is important to note, that the first objective, a common metric of comparison in this area Gómez-Bombarelli et al. (2016b); Kusner et al. (2017); Jin et al. (2018), is mis-specified. From a chemical standpoint it is undesirable to maximize the logP score as is being done here. Rather it is preferable to optimize logP to be in a range that is in accordance with the Lipinski Rule of Five Lipinski et al. (1997). We use the penalized logP objective here because regardless of its relevance for chemistry, it serves as a point of comparison against other methods.

## 2.4 CONSTRAINED BAYESIAN OPTIMIZATION OF MOLECULES

We now describe our extension to the Bayesian Optimization procedure followed by Gómez-Bombarelli et al. (2016b). Expressed formally, the constrained optimization problem is

$$\max_m f(m) \text{ s.t. } \Pr(\mathcal{C}(m)) \geq 1 - \delta$$

where $f(m)$ is a black-box objective function, $\Pr(\mathcal{C}(m))$ denotes the probability that a boolean constraint $\mathcal{C}(m)$ is satisfied and $1 - \delta$ is some user-specified minimum confidence that the constraint is satisfied (Gelbart et al., 2014). The constraint is that a latent point must decode successfully a large fraction of the times decoding is attempted. The black-box objective function is noisy because a single latent point may decode to multiple molecules when the model makes a mistake, obtaining different values under the objective. In practice, $f(m)$ is one of the objectives described in section 2.3

## 2.5 EXPECTED IMPROVEMENT WITH CONSTRAINTS (EIC)

EIC may be thought of as expected improvement (EI),

$$\text{EI}(m) = \mathbb{E}_{f(m)}\big[\max(0, \eta - f(m))\big],$$

that offers improvement only when a set of constraints are satisfied (Schonlau et al., 1998):

$$\text{EIC}(m) = \text{EI}(m) \Pr(\mathcal{C}(m)).$$

The incumbent solution $\eta$ in $\text{EI}(m)$, may be set in an analogous way to vanilla expected improvement (Gelbart, 2015) as either:

1. The best observation in which all constraints are observed to be satisfied.
2. The minimum of the posterior mean such that all constraints are satisfied.

The latter approach is adopted for the experiments performed in this paper. If at the stage in the Bayesian optimization procedure where a feasible point has yet to be located, the form of acquisition function used is that defined by (Gelbart, 2015)

$$
\text{EIC}(m) = \left\{ \begin{array}{ll} \Pr(\mathcal{C}(m)\text{EI}(m), & \text{if } \exists m, \Pr(\mathcal{C}(m)) \geq 1 - \delta \\ \Pr(\mathcal{C}(m)), & \text{otherwise} \end{array} \right.
$$

with the intuition being that if the probabilistic constraint is violated everywhere, the acquisition function selects the point having the highest probability of lying within the feasible region. The algorithm ignores the objective until it has located the feasible region. It would also have been possible to adopt the methodologies of Rainforth et al. (2016); Mueller et al. (2017) under the assumption that the constraint is cheap to evaluate.

## 3    RELATED WORK

Gómez-Bombarelli et al. (2016b); Segler et al. (2017) constructed character-level generative models of SMILES strings using recurrent decoders. These models both suffered from the problem that they produced a high proportion of invalid SMILES strings. Blaschke et al. (2017) compared a range of autoencoder architectures in terms of their ability to produce valid molecules. Kusner et al. (2017); Dai et al. (2018) addressed the issue of generating valid molecules explicitly by imposing syntactic and semantic constraints using context free and attribute grammars while Janz et al. (2017); Guimaraes et al. (2017) leveraged additional training signal to enforce molecular validity.

A potential drawback of these methods is that the constraints may favour the generation of simple molecules. Simonovsky & Komodakis (2018); Li et al. (2018) build generative models of molecular graphs but do not impose any constraints for molecular validity. Jin et al. (2018) build a generative model of molecular graphs with constraints in order to achieve 100% validity. One possible disadvantage of this approach is that each molecule is assumed to be constructed from a fixed vocabulary of subgraphs and so it will be impossible to generate a molecule comprised of subgraphs outside this vocabulary. De Cao & Kipf (2018) use a graph-based generative model which enforces a quality constraint in the latent space. Direct comparison with the aforementioned methods is complicated due to the orthogonality of the assumptions made for each model. As such, we compare out method against Gómez-Bombarelli et al. (2016b) only.

Our principle contribution is in showing that valid molecules can be generated without encoding assumptions into the design model that limit the complexity and diversity of the generated molecules.

## 4    EXPERIMENT I: DRUG DESIGN

In this section we conduct an empirical test of the hypothesis from (Gómez-Bombarelli et al., 2016b) that the decoder's lack of efficiency is due to data point collection in "dead regions" of the latent space far from the data on which the VAE was trained. We use this information to construct a binary classification Bayesian Neural Network (BNN) to serve as a constraint function that outputs the probability of a latent point being valid. Secondly, we compare the performance of our constrained Bayesian optimization implementation against the original model (baseline) in terms of the numbers of valid and drug-like molecules generated. Thirdly, we compare the quality of the molecules produced by constrained Bayesian optimization with those of the baseline model.

### 4.1    IMPLEMENTATION

The implementation details of the encoder-decoder network as well as the sparse GP for modelling the objective remain unchanged from (Gómez-Bombarelli et al., 2016b). For the constrained Bayesian optimization algorithm, the BNN is constructed with 2 hidden layers each 100 units wide with ReLU activation functions and a logistic output. Minibatch size is set to 1000 and the network is trained for 5 epochs with a learning rate of 0.0005. 20 iterations of parallel Bayesian optimization are performed using the Kriging-Believer algorithm in all cases. Data is collected in batch sizes of 50. The same training set as (Gómez-Bombarelli et al., 2016b) is used, namely $249, 456$ drug-like molecules drawn at random from the ZINC database (Irwin et al., 2012).

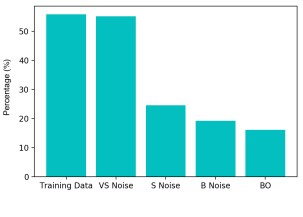 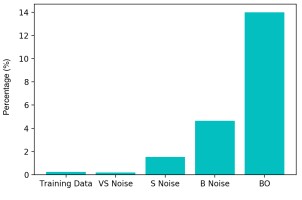 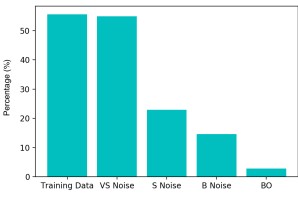

(a) % Valid Molecules      (b) % Methane Molecules      (c) % Drug-like Molecules

Figure 3: Experiments on 5 disjoint sets comprising 50 latent points each. Very small (VS) Noise are training data latent points with approximately 1% noise added to their values, Small (S) Noise have 10% noise added to their values and Big (B) Noise have 50% noise added to their values. All latent points underwent 500 decode attempts and the results are averaged over the 50 points in each set. The percentage of decodings to: **a)** valid molecules **b)** methane molecules. **c)** drug-like molecules.

## 4.2 Diagnostic Experiments and Labelling Criteria

These experiments were designed to test the hypothesis that points collected by Bayesian optimization lie far away from the training data in latent space. In doing so, they also serve as labelling criteria for the data collected to train the BNN acting as the constraint function. The resulting observations are summarized in Figure 3.

There is a noticeable decrease in the percentage of valid molecules decoded as one moves further away from the training data in latent space. Points collected by Bayesian optimization do the worst in terms of the percentage of valid decodings. This would suggest that these points lie farthest from the training data. The decoder over-generates methane molecules when far away from the data. One hypothesis for why this is the case is that methane is represented as 'C' in the SMILES syntax and is by far the most common character. Hence far away from the training data, combinations such as 'C' followed by a stop character may have high probability under the distribution over sequences learned by the decoder.

Given that methane has far too low a molecular weight to be a suitable drug candidate, a third plot in Figure 3(c), shows the percentage of decoded molecules such that the molecules are both valid and have a tangible molecular weight. The definition of a tangible molecular weight was interpreted somewhat arbitrarily as a SMILES length of 5 or greater. Henceforth, molecules that are both valid and have a SMILES length greater than 5 will be loosely referred to as drug-like. This is not to imply that a molecule comprising five SMILES characters is likely to be drug-like, but rather this SMILES length serves the purpose of determining whether the decoder has been successful or not.

As a result of these diagnostic experiments, it was decided that the criteria for labelling latent points to initialize the binary classification neural network for the constraint would be the following: if the latent point decodes into drug-like molecules in more than 20% of decode attempts, it should be classified as drug-like and non drug-like otherwise.

## 4.3 Molecular Validity

The BNN for the constraint was initialized with $117,440$ positive class points and $117,440$ negative class points. The positive points were obtained by running the training data through the decoder assigning them positive labels if they satisfied the criteria outlined in the previous section. The negative class points were collected by decoding points sampled uniformly at random across the 56 latent dimensions of the design space. Each latent point undergoes 100 decode attempts and the most probable SMILES string is retained. $J_{\text{comp}}^{\text{logP}}(m)$ is the choice of objective function. The relative performance of constrained Bayesian optimization and unconstrained Bayesian optimization (baseline) (Gómez-Bombarelli et al., 2016b) is compared in 4(a).

The results show that greater than 80% of the latent points decoded by constrained Bayesian optimization produce drug-like molecules compared to less than 5% for unconstrained Bayesian optimization. One must account however, for the fact that the constrained approach may be decoding

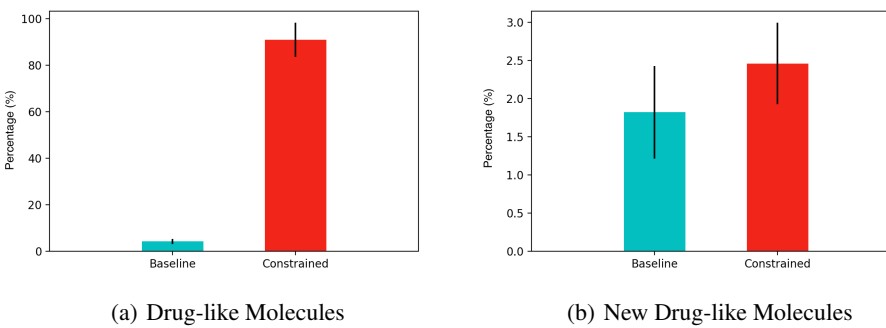

(a) Drug-like Molecules

(b) New Drug-like Molecules

Figure 4: **a)** The percentage of latent points decoded to drug-like molecules. The results are from 20 iterations of Bayesian optimization with batches of 50 data points collected at each iteration (1000 latent points decoded in total). The standard error is given for 5 separate train/test set splits of 90/10.

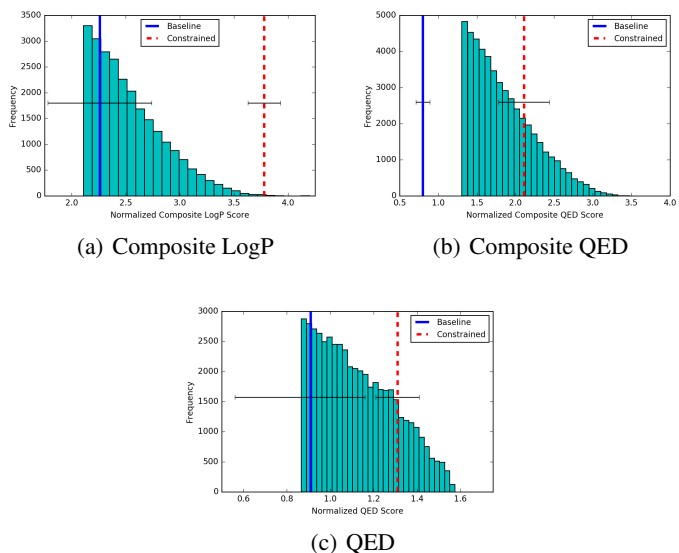

(a) Composite LogP

(b) Composite QED

(c) QED

Figure 5: The best scores for new molecules generated from the baseline model (blue) and the model with constrained Bayesian optimization (red). The vertical lines show the best scores averaged over 5 separate train/test splits of 90/10. For reference, the histograms are presented against the backdrop of the top 10% of the training data in the case of Composite LogP and QED, and the top 20% of the training data in the case of Composite QED.

multiple instances of the same novel molecules. Constrained and unconstrained Bayesian optimization are compared on the metric of the percentage of unique novel molecules produced in 4(b).

One may observe that constrained Bayesian optimization outperforms unconstrained Bayesian optimization in terms of the generation of unique molecules, but not by a large margin. A manual inspection of the SMILES strings collected by the unconstrained optimization approach showed that there were many strings with lengths marginally larger than the cutoff point, which is suggestive of partially decoded molecules. As such, a fairer metric for comparison should be the quality of the new molecules produced as judged by the scores from the black-box objective function. This is examined next.

Table 1: Percentile of the averaged new molecule score relative to the training data. The results of 5 separate train/test set splits of 90/10 are provided.

| OBJECTIVE | BASELINE | CONSTRAINED |
|---|---|---|
| LogP Composite | $36\pm 14$ | $92\pm 4$ |
| QED Composite | $14\pm 3$ | $72\pm 10$ |
| QED | $11\pm 2$ | $79\pm 4$ |

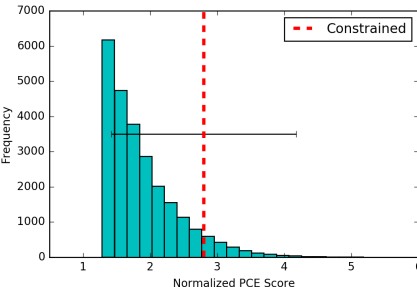

Figure 6: The best scores for novel molecules generated by the constrained Bayesian optimization model optimizing for PCE. The results are averaged over 3 separate runs with train/test splits of 90/10.

## 4.4 MOLECULAR QUALITY

The results of Figure 5 indicate that constrained Bayesian optimization is able to generate higher quality molecules relative to unconstrained Bayesian optimization across the three drug-likeness metrics introduced in section 2.3. Over the 5 independent runs, the constrained optimization procedure in every run produced new drug-like molecules ranked in the 100th percentile of the distribution over training set scores for the $J_{\text{comp}}^{\text{logP}}(m)$ objective and over the 90th percentile for the remaining objectives. Table 1 gives the percentile that the averaged score of the new molecules found by each process occupies in the distribution over training set scores.

## 5 EXPERIMENT II: MATERIAL DESIGN

In order to show that the constrained Bayesian optimization approach is extensible beyond the realm of drug design, we trained the model on data from the Harvard Clean Energy Project (Hachmann et al., 2011; 2014) to generate molecules optimized for power conversion efficiency (PCE).

## 5.1 IMPLEMENTATION

A neural network was trained to predict the PCE of $200,000$ molecules drawn at random from the Harvard Clean Energy Project dataset using 512-bit Morgan circular fingerprints Rogers & Hahn (2010) as input features with bond radius of 2 using RDKit. If unmentioned the details of the implementation remain the same as section 4.

## 5.2 RESULTS

The results are given in Figure 6. The averaged score of the new molecules generated lies above the 90th percentile in the distribution over training set scores. Given that the objective function in this instance was learned using a neural network, advances in predicting chemical properties from data Duvenaud et al. (2015); Ramsundar et al. (2015) are liable to yield concomitant improvements in the optimized molecules generated through this approach.

## 6 Conclusion and Future Work

### 6.1 Contributions

The reformulation of the search procedure in the Automatic Chemical Design model as a constrained Bayesian optimization problem has led to concrete improvements on two fronts:

1. Validity - The number of valid molecules produced by the constrained optimization procedure offers a marked improvement over the original model. Notably, by applying constraints solely to the latent space of the variational autoencoder, the method does not require model assumptions that compromise either the complexity or the diversity of the generated molecules.

2. Quality - For five independent train/test splits, the scores of the best molecules generated by the constrained optimization procedure consistently ranked above the 90th percentile of the distribution over training set scores for all objectives considered. The ability to find new molecules that are competitive with the very best of a training set of already drug-like molecules is a powerful demonstration of the model's capabilities. As a further point, the generality of the approach should be emphasised. This approach is liable to work for a large range of objectives encoding countless desirable molecular properties.

### 6.2 Future Work

Future work could investigate whether performance gains can be achieved through the implementation of a more accurate constraint model. Recent work by Blaschke et al. (2017); Polykovskiy et al. (2018); Tabor et al. (2018); Aumentado-Armstrong (2018); Sanchez-Lengeling & Aspuru-Guzik (2018) has featured a more targeted search for novel compounds. This represents a move towards more industrially-relevant objective functions for Bayesian Optimization which should ultimately replace the chemically mis-specified objectives, such as the penalized logP score, identified here.

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

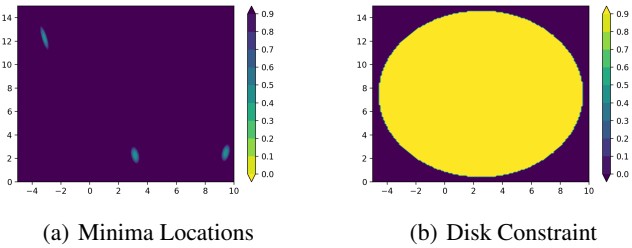

(a) Minima Locations        (b) Disk Constraint

Figure 7: Constrained Bayesian optimization of the 2D Branin-Hoo Function.

## A   Toy Experiment: The Branin-Hoo Function

The Branin-Hoo function will act as a toy problem on which to test the functionality of the algorithmic implementation for constrained Bayesian optimization. The particular variant of the Branin-Hoo optimization of interest here is the constrained formulation of the problem as featured in Gelbart et al. (2014). This Branin-Hoo function has three global minima at the coordinates $(-\pi, 12.275), (\pi, 2.275)$ and $(9.42478, 2.475)$. In order to formulate the problem as a constrained optimization problem, a disk constraint on the region of feasible solutions is introduced. In contrast to the formulation of the problem in Gelbart et al. (2014), the disk constraint is coupled in this scenario in the sense that the objective and the constraint will be evaluated jointly at each iteration of Bayesian optimization. In addition, the observations of the black-box objective function will be assumed to be noise-free. The minima of the Branin-Hoo function as well as the disk constraint are illustrated in Figure 7.

The disk constraint eliminates the upper-left and lower-right solutions, leaving a unique global minimum at $(\pi, 2.275)$. Given that our implementation of constrained Bayesian optimization relies on the use of a sparse GP as the underlying statistical model of the black-box objective and as such is designed for scale as opposed to performance, the results will not be compared directly against those of Gelbart et al. (2014) who use a full GP to model the objective. It will be sufficient to compare the performance of the algorithm against random sampling. Both the sequential Bayesian optimization algorithm and the parallel implementation using the Kriging-Believer algorithm Ginsbourger et al. (2010) are tested.

### A.1   Implementation

A Sparse GP featuring the FITC approximation, based on the implementation of Bui et al. (2016) is used to model the black-box objective function. The kernel choice is exponentiated quadratic with ARD lengthscales. The number of inducing points $M$ was chosen to be 20 in the case of sequential Bayesian optimization, and 5 in the case of parallel Bayesian optimization using the Kriging-Believer algorithm. The sparse GP is trained for 400 epochs using Adam Kingma & Ba (2014) with the default parameters and a learning rate of 0.005. The minibatch size is chosen to be 5. The extent of jitter is chosen to be 0.00001. A Bayesian Neural Network (BNN), adapted from the MNIST digit classification network of Hernández-Lobato et al. (2016) is trained using black-box alpha divergence minimization to model the constraint.

The network has a single hidden layer with 50 hidden units, Gaussian activation functions and logistic output units. The mean parameters of $q$, the approximation to the true posterior, are initialized by sampling from a zero-mean Gaussian with variance $\frac{2}{d_{\text{in}}+d_{\text{out}}}$ according to the method of Glorot & Bengio (2010), where $d_{\text{in}}$ is the dimension of the previous layer in the network and $d_{\text{out}}$ is the dimension of the next layer in the network. The value of $\alpha$ is taken to be 0.5, minibatch sizes are taken to be 10 and 50 Monte Carlo samples are used to approximate the expectations with respect to $q$ in each minibatch. The BNN adapted from Hernández-Lobato et al. (2016) was implemented in the Theano library Theano Development Team (2016). The LBFGs method Liu & Nocedal (1989) was used to optimize the EIC acquisition function in all experiments.

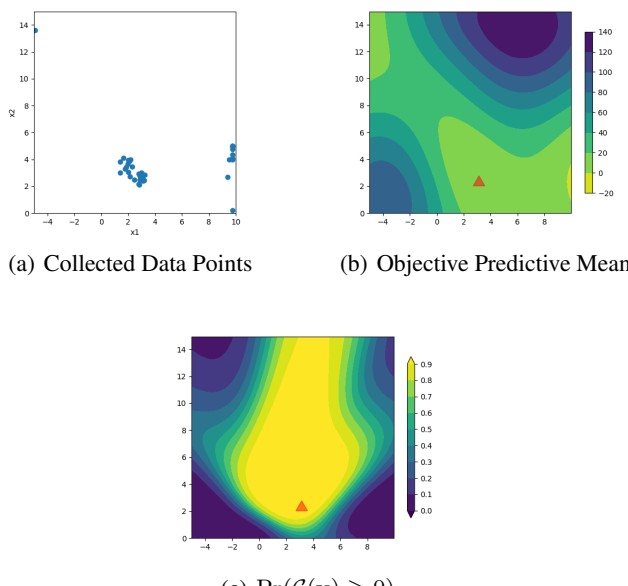

(a) Collected Data Points      (b) Objective Predictive Mean

(c) $\Pr(\mathcal{C}(\mathbf{x}) \geq 0)$

Figure 8: **a)** Data points collected over 40 iterations of sequential Bayesian optimization. **b)** Contour plot of the predictive mean of the sparse GP used to model the objective function. Lighter colours indicate lower values of the objective. **c)** The contour learned by the BNN giving the probability of constraint satisfaction.

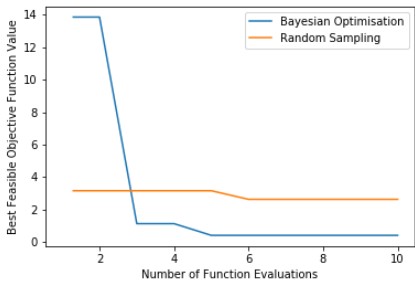

Figure 9: Performance of Parallel Bayesian Optimization with EIC against Random Sampling.

## A.2   RESULTS

The results of the sequential constrained Bayesian optimization algorithm with EIC are shown in Figure 8. The algorithm was initialized with $50$ labeled data points drawn uniformly at random from the grid depicted. $40$ iterations of Bayesian optimization were carried out.

The figures show that the algorithm is correctly managing to collect data in the vicinity of the single feasible minimum. Figure 9 compares the performance of parallel Bayesian optimization using the Kriging-Believer algorithm against the results of random sampling. Both algorithms were initialized using $10$ data points drawn uniformly at random from the grid on which the Branin-Hoo function is defined and were run for $10$ iterations of Bayesian optimization. At each iteration a batch of $5$ data points was collected for evaluation.

After $10$ iterations, the minimum feasible value of the objective function was $0.42$ for parallel Bayesian optimization with EIC using the Kriging-Believer algorithm and $2.63$ for random sampling. The true minimum feasible value is $0.40$.

It is very likely that the parallel constrained Bayesian optimization algorithm experienced a serendipitous initialisation on the single run shown.

## A.3 DISCUSSION

The Branin-Hoo experiment is designed to yield some visual intuition for the constrained Bayesian Optimization implementation in two dimensions before moving to higher dimensional molecular space. The results demonstrate that the implementation of constrained Bayesian optimization is behaving as expected in so far as the constraint in the problem is recognized and the search procedure outperforms random sampling.

It could be worth performing some investigation into how much worse the sparse GP performs relative to the full GP in the constrained setting. Another aspect that could be explored is the impact of the initialization.

