# OpenReview forum: "Constrained Bayesian Optimization for Automatic Chemical Design"
_ICLR.cc/2019/Conference_

### Official Review · AnonReviewer3 · 2018-10-20
**lack of empirical evidence on being able to do sequential selections**

**Rating:** 5
**Confidence:** 4

**Review:**

The authors proposed a new method improving a previous Bayesian optimization approach for chemical design (Gomez-Bombarelli et al., 2016b) by addressing the problem that data points need to have valid molecular structures. The main contribution is a constrained Bayesian optimization approach that take into account the constraint on the probability of being valid.

My biggest concern of this paper is that it is not really using sequential evaluations to do automatic design of experiments on molecules. The experiments do not seem to fully support the proposed approach in terms of being able to adaptively do sequential evaluations.

Detailed comments:
1. The term inside the expectation of the EI criterion in Sec 2.5 should be max(0, f(m)-\eta) rather than max(0, \eta - f(m)).
2. The EIC criterion the authors adopted uses Pr(C(m)) if the constraint is violated everywhere with high probability. It seems like Pr(C(m)) does not have the ability to explore regions with high uncertainty. How does this approach compare to Bayesian level set estimation approaches like
B. Bryan, R. C. Nichol, C. R. Genovese, J. Schneider, C. J. Miller, and L. Wasserman, “Active learning for identifying function threshold boundaries,” in NIPS, 2006
I. Bogunovic, J. Scarlett, A. Krause, and V. Cevher, “Truncated variance reduction: A unified approach to bayesian optimization and level-set estimation,” in NIPS, 2016.
3. It would be good to explain in more detail how a constraint is labeled to be valid or invalid.
4. What is the space of m in Sec 2.3 and the space of m in Sec 2.4? It seems like there is a discrepancy: Sec 2.3 is talking about m as a molecule but Sec 2.4 describes f as a function on the latent variable? It would be good to be clear about it.
5. In the experiment, I think it is very necessary to show the effectiveness of the constrained BO approach in terms of how the performance varies as a function of the number of evaluations on the constraint and the function. The current empirical results only support the claim that the constrained BO approach is able to output more valid latent variables and the function values from constrained BO is higher than vanilla BO under the same number of training data. It is also strange why there is a set of test data.


Typos/format:
1. citation format problems across the paper, e.g.
"Gomez-Bombarelli et al. (Gomez-Bombarelli et al., 2016b) presented Automatic Chem"
"SMILES strings Weininger (1988) are a means of representing molecules as a character sequence."
It's likely a problem of misuse of \cite, \citep.
2. no period at the end of Sec 2.4

---

### Official Review · AnonReviewer2 · 2018-10-26
**Novelty is limited. No comparison with SOTA models**

**Rating:** 4
**Confidence:** 3

**Review:**

This paper proposes to improve the chemical compound generation by the Bayes optimization strategy, not by the new models.
The main proposal is to use the acquisition that switches the function based on the violation of a constraint, estimated via a BNN.

I understand that the objective function, J_{comp}^{QED} is newly developed by the authors, but not intensively examined in the experiments.
The EIC, and the switching acquisition function is developed by (Schonlau+ 1998; Gelbard, 2015).
So I judge the technical novelty is somewhat limited.

It is unfortunate that the paper lacks intensive experimental comparisons with "model assumption approaches".
My concern is that the baseline is rely on the SMILE strings.
It is well known that the string-based generators are much weaker than the graph-based generators.
In fact, the baseline model is referred as "CVAE" in (Jing+, 2018) and showed very low scores against other models.

Thus, we cannot tell that these graph-based, "model-assumption" approaches are truly degraded in terms of the validity and the variety of generated molecules,
compared to those generated by the proposed method.
In that sense, preferable experimental setting is that to
test whether the constrained Bayesian optimization can boost the performance of the graph-based SOTA models.


+ Showing that we can improve the validity the modification of the acquisition functions
- Technical novelty is limited.
- No comparison with SOTA models in "graph-based, model assumption approaches".

---

### Official Review · AnonReviewer1 · 2018-11-03
**Insufficient presentations and evidences to conclude the improvement**

**Rating:** 3
**Confidence:** 4

**Review:**

Summary:
This paper proposes a novel method for generating novel molecules with some targeted properties. Many studies on how to generate chemically valid molecular graphs have been done, but it is still an open problem due to the essential difficulty of generating discrete structures from any continuous latent space. From this motivation, the 'constrained' Bayesian optimization (BO) is applied and analyzed. Posing 'constraints' on the validity is realized by probability-weighting onto the expected improvement scores in BO. The 'validity' probability is learned beforehand by Bayesian neural nets in a supervised way. As empirical evaluations, two case studies are presented, and quality improvements of generated molecules are observed.

Comment:
- The presentation would be too plain to find what parts are novel contributions. Every part of presentations seems originated from some past studies at the first glance.

- In this paper, how to pose the validity 'constraint' onto Bayesian optimization would be the main concern. Thus if it is acquired through supervised learning of Bayesian neural nets in advance, that part should be explained more in details. How do we collect or setup the training data for that part? Is it valid to apply such trained models to the probability weighting P(C(m)) on EI criteria in the test phase? Any information leakage does not happen?

- The implementations of constrained BO is just directly borrowed from Gelbart, 2015 including parallel BO with kriging believer heuristics? The description on the method is totally omitted and would need to be included.

- How training of Bayesian neural nets for 'Experiment II' are performed? What training datasets are used? Is it the same as those for 'Experiment I' even though the target and problem are very different?

Pros:
- a constrained Bayesian optimization with weighing EI by the probabilities from pre-trained Bayesian neural nets applied to the hot topic of valid molecule generations.
- Experiments observe the quality improvements

Cons:
- unclear and insufficient descriptions of the method and the problem
- novel contributions are unclear

---

### Meta-Review · Area_Chair1 · 2018-12-14
**clear rejection; no rebuttal**

**Confidence:** 5
**Recommendation:** Reject

**Metareview:**

This paper proposes to use constrained Bayesian optimization to improve the chemical compound generation. Unfortunately, the reviewers raises a range of critical issues which are not responded by authors' rebuttal.